# Meta-Analysis of Modulated Electro-Hyperthermia and Tumor Treating Fields in the Treatment of Glioblastomas

**DOI:** 10.3390/cancers15030880

**Published:** 2023-01-31

**Authors:** Attila Marcell Szasz, Elisabeth Estefanía Arrojo Alvarez, Giammaria Fiorentini, Magdolna Herold, Zoltan Herold, Donatella Sarti, Magdolna Dank

**Affiliations:** 1Division of Oncology, Department of Internal Medicine and Oncology, Semmelweis University, 1083 Budapest, Hungary; 2Oncología Radioterápica, Servicios y Unidades Asistenciales, Hospital Universitario Marqués de Valdecilla, 39008 Santander, Spain; 3Medical Institute of Advanced Oncology, 28037 Madrid, Spain; 4Department of Oncology, Azienda Ospedaliera “Ospedali Riuniti Marche Nord”, 61121 Pesaro, Italy; 5IHF Integrative Oncology Outpatient Clinic, 40121 Bologna, Italy; 6Department of Internal Medicine and Hematology, Semmelweis University, 1088 Budapest, Hungary

**Keywords:** astrocytoma, glioblastoma, modulated electro-hyperthermia, tumor treating fields

## Abstract

**Simple Summary:**

Glioblastoma is a highly aggressive brain tumor, which has a very poor 5-year survival rate (<5%). In the last decades, the concomitant use of two non-invasive, electromagnetic devices, modulated electro-hyperthermia (mEHT) and Tumor Treating Fields (TTF) has been introduced. Both mEHT and TTF have specific anti-tumor effects, which can help to achieve a more efficient treatment of patients and a higher rate of therapeutic response. In this meta-analysis we investigated how patient survival rates change if either device is used. The significant difference in the 1-year survival rates between the treated (>60%) and untreated groups (historical data: <40%) confirms the observation that the use of both mEHT and TTF in the treatment of glioblastomas benefits patients. In addition, it is important to emphasize that most studies have proven that the mEHT or TTF-treated patients’ quality of life is much better than that of the untreated patients.

**Abstract:**

Background: Glioblastoma is one of the most difficult to treat and most aggressive brain tumors, having a poor survival rate. The use of non-invasive modulated electro-hyperthermia (mEHT) and Tumor Treating Fields (TTF) devices has been introduced in the last few decades, both of which having proven anti-tumor effects. Methods: A meta-analysis of randomized and observational studies about mEHT and TTF was conducted. Results: A total of seven and fourteen studies about mEHT and TTF were included, with a total number of 450 and 1309 cases, respectively. A 42% [95% confidence interval (95% CI): 25–59%] 1-year survival rate was found for mEHT, which was raised to 61% (95% CI: 32–89%) if only the studies conducted after 2008 were investigated. In the case of TTF, 1-year survival was 67% (95% CI: 53–81%). Subgroup analyses revealed that newly diagnosed patients might get extra benefits from the early introduction of the devices (mEHT all studies: 73% vs. 37%, *p* = 0.0021; mEHT studies after 2008: 73% vs. 54%, *p* = 0.4214; TTF studies: 83% vs. 52%, *p* = 0.0083), compared with recurrent glioblastoma. Conclusions: Our meta-analysis showed that both mEHT and TTF can improve glioblastoma survival, and the most benefit may be achieved in newly diagnosed cases.

## 1. Introduction

Based on the 2020 GLOBOCAN report, more than 300,000 new central nervous system tumors are confirmed each year, with more than 250,000 deaths [1,2]. Among these, gliomas are the most common [3], which have different origins (e.g., astrocytes and oligodendrocytes) [4,5]. As per the latest WHO classification (2021), depending on the origin and the mutation of isocitrate dehydrogenase (IDH), three major types of diffuse gliomas are known: astrocytomas (IDH mutant), oligodendroglioma (IDH mutant and 1p/19q co-deleted) and glioblastomas (IDH wild type) [6]. Glioblastomas are known to be the most aggressive and the most occurring type [6,7]. Up to 60% of all malignant primary brain tumors in adults are estimated to be glioblastomas [8]. For decades, the only treatment options for astrocytoma/glioblastoma patients were surgery and radiotherapy [9], but with the introduction of concurrent and/or adjuvant temozolomide chemotherapy patient survival significantly improved [10]. Thanks to the combined effect of temozolomide and radiotherapy, 1-year survival has improved to 30–40%, and some studies reported even higher ones (>80%) [11], however, only modest median overall survivals can usually be achieved [12,13].

In the last two decades, semi-invasive and non-invasive electromagnetic devices/techniques with anti-tumoral effects have been introduced that can be used concomitantly and/or palliatively in the treatment of glioblastomas to supplement chemoradiotherapy. Magnetic hyperthermia, in which the local deposition of magnetic nanoparticles is needed prior the application of an external alternating magnetic field, belongs to the former type [14]. In contrast, modulated electro-hyperthermia (mEHT) and Tumor Treating Fields (TTF) are non-invasive techniques; the devices only have be placed on the skin of the patients. In this article, the latter type is presented in more detail. The most optimal is when the mEHT treatment is done three-times a week, while TTF has to be worn for >18 h daily [15,16,17,18,19,20,21,22,23]. With an optimal frequency of 200 kHz, TTF focuses on nonthermal effects on the cytokinetic “neck” using capacitive coupling [24,25]. The electric field of TTF reorients the high polarizable microtubules and actin fibers, and it may arrest the cytoskeleton’s polymerization process and inhibit the assembling of the mitotic spindle, ultimately blocking the finalization of the last phase of cell division and thus inhibiting further proliferation [26]. TTF also stimulates macrophages, promoting immunogenic cell death via dendritic cell recruitment and maturation, reducing the capacity of cancer cells for migration and invasion, preventing the inhibitory effects of the PI3K/Akt/mTORC1 signaling pathway on autophagy and increasing DNA replication stress and double-strand break formation [27]. Moreover, the electric field generated by TTF increases membrane permeability enhancing the effect of chemotherapy significantly [28]. It has to be noted though, this latter effect is reversible [28], therefore, it can be expected that the improved chemo-sensitivity will probably reduce when TTF is not in use.

In contrast, mEHT accurately balances both the nonthermal electric processes and the low-power thermal effects. It operates in a precision capacitive coupled impedance matched way, working on a radiofrequency of 13.56 MHz [29]. mEHT exploits various biophysical differences of cancer cells. For example, energy absorption on the membrane rafts is different than those of healthy host cells, and damage-associated molecular patterns (DAMPS) will also occur. All of these eventually lead to programmed or immunogenic tumor cell death [30,31,32]. It has also been reported that mEHT can enhance DNA fragmentation of tumor cells, increase the fraction of cells with low mitochondrial membrane potential, increase the concentration of intracellular Ca^2+^, increase the Fas, c-Jun N-terminal kinases and MAPK/ERK signaling pathways, increase the expression of pro-apoptotic Bcl-2 family proteins and can up-regulate the expression of genes associated with the molecular function of cell death (*EGR1*, *JUN*, and *CDKN1A*) and silencing others associated with cytoprotective functions [33,34]. It also has to be mentioned that the use of mEHT is also feasible in tumors of other locations as well [34].

Although both mEHT and TTF have advantageous effects against cancer cells, their use in routine oncology still awaits. The general acceptance of TTF—and perhaps also the awareness about it—is wider than that of the mEHT. One of the latest developments in the widespread application of TTF is that the American Society of Clinical Oncology (ASCO) guidelines recommend TTF therapy for newly diagnosed supratentorial glioblastoma without isocitrate dehydrogenase mutations after the completion of chemoradiation therapy [9]. In contrast, only clinical trial results and development reports are available for mEHT. Therefore, the main purpose of this meta-analysis is to provide comprehensive data and a systematic literature review on the clinical importance of mEHT in glioblastoma. Moreover, presenting the same information about TTF and, last but not least, the direct comparison of the two devices were further goals of this study.

## 2. Materials and Methods

### 2.1. Search Strategy

The study was conducted following the Preferred Reporting Items for Systematic Reviews and Meta-Analyses (PRISMA) guidelines [35]. Ethical approval was not required for the study due to the fact that the article presents aggregate data from previously published studies. The meta-analysis was registered in the PROSPERO database with the Registration Number: CRD42022385535. The search for eligible publications was performed in the BioMed Central (BMC), ClinicalTrials.gov, Cochrane Library, European Union Drug Regulating Authorities Clinical Trials Database (EudraCT), Excerpta Medica Database (Embase), PubMed—Medical Literature Analysis and Retrieval System Online (MEDLINE), World Health Organization’s International Clinical Trials Registry Platform and in the University hospital Medical Information Network (UMIN) Clinical Trials Registry (for Japan) databases from their inception to 30 June 2022. The following search strings were used. The terms “glioblastoma” and “glioma” were combined with “electrohyperthermia”, “electro hyperthermia”, “electro-hyperthermia”, “hyperthermia”, “modulated electrohyperthermia”, “oncotherm”, “oncothermia”, “alternating electric fields”, “TTFields”, “tumor treating fields” and “tumor-treating fields” using the logical operator AND. Furthermore, individual searches for “EHY-2030”, “EHY 2030” and “EHY2030” were also performed. Language restrictions were not used.

Inclusion criteria for the studies were to contain survival data, either in the form of x-year survival rates, the number of deaths in x-year or survival curves from which the proportion of patients alive at the specific timepoints can be read. Concomitant or monotherapeutic use of mEHT/TTF was no limiting factor for inclusion. Exclusion criteria included if the observation period of the study was shorter than 1 year, if the patients treated with one of the two devices could not be separated from the controls (mixed study groups), or if the study contained only median survival and/or hazard rates only.

Type of publications (review, conference abstract, etc.) and species information (human vs. other) were retrieved from databases, where available. Publications belonging to the following categories were excluded without further review: reviews, conference abstracts, in vitro and animal studies and theoretical works. No automation tool was used during the literature search. The literature search was conducted independently by two investigators (A.M.S. and Z.H.), and any discrepancies were resolved by consensus and, if necessary, by the opinion of a third reviewer (M.D.).

### 2.2. Data Extraction

Collected data included general information about the study: name of author(s) and year of publication. The following study characteristics were recorded from each publication: type of study [prospective observational study, retrospective observational study or randomized clinical trial (RCT)], sample size of treated and/or control groups, median age of patients, percentage of females, 1-year, 2-year and/or 3-year survival rate if available, and newly diagnosed or recurrent tumor. If the authors did not directly present the x-year survival rate but the corresponding survival curve(s) of the cohort(s) was drawn, the percentage of patients alive at the specific timepoints was read from the survival curve(s). It has to be noted, that grade 4 astrocytomas were previously termed as IDH-mutant glioblastomas, compared with the current WHO classification [6], and most of the articles used for the meta-analysis are older than the current classification, the differences in their nomenclature arise from this.

### 2.3. Statistical Analysis

Statistical analyses were performed within the R for Windows version 4.2.1 environment (R Foundation for Statistical Computing, 2022, Vienna, Austria) using the R package *meta* (version 6.0-0) [36]. Survival rate at specific timepoints (1-year, 2-year or 3-year, if available) was used for the effect size measure and random-effects models were performed. To estimate the heterogeneity variance measure (τ^2^) the restricted maximum-likelihood method [37] was applied with the Q profile method for confidence interval [38]. Between-study heterogeneity was described by the Higgin’s and Thompson’s *I*^2^ statistic [39], and publication bias was tested using the Egger’s regression test [40]. The Mantel–Haenszel method was used for group comparisons [41,42], plural models (fixed-effects model between subgroups but studies within the subgroups are pooled using the random-effects model) were used for subgroup analyses [38] and meta-regression methods were used to assess possible confounding/biasing effects (e.g., publication year) over effect size [38]. Forest plots were used to graphically represent study results.

## 3. Results

### 3.1. Studies Investigating Modulated Electro-Hyperthermia in Glioblastoma

The electronic database searches for studies about mEHT in glioblastoma patients resulted in a total of 2586 articles. After the removal of duplicates, non-human studies including animal and cellular research reports, reviews and meeting/conference abstracts, 686 articles remained for title and abstract screening. In total, 650 articles were excluded because they either reported results from other tumors, presented results from an animal and/or cellular experiment, were unavailable or had different study interests than the current meta-analysis (e.g., health-economy). Thirty-six studies were considered for full text assessment; however, twenty-seven further studies needed to be excluded. Of the remaining nine studies, two–two articles belonged to the same work ([43,44,45,46]), of which only one–one paper ([43,46]) was used for the meta-analysis, resulting in a total of seven available full text articles to be included in the analyses (Figure 1).

Details of the seven mEHT studies [43,46,47,48,49,50,51] selected for analysis can be read in Table 1. Six [43,46,47,48,49,50] and one [51] studies investigated the effect of mEHT in recurrent/late stage and in newly diagnosed glioblastoma, respectively. A comparison between mEHT-treated and control patients was only present only in one study [46]. The total number of patients included in the meta-analysis was 450, of whom 292 (64.9%) died during the first year after study inclusion. A 42.33% 1-year survival rate [95% confidence interval (CI): 25.17–59.49%] was estimated (Figure 2). Although the heterogeneity between studies was high [91.3% (95% CI: 84.6–95.1%)], no publication bias was present based on the results of the Egger’s regression test (*p* = 0.6449).

Further analysis was performed to elucidate the confounding effects behind high heterogeneity. In total, 72.26% of the difference in the true effect sizes could be explained by the publication year (*p* = 0.0008). Comparing the studies published before and after 2008, it was found that in early studies the 1-year survival rate was 31.22% (95% CI: 24.81–37.62%), while in the ones after 2008 it was 60.63% (95% CI: 32.21–89.05%; *p* = 0.0478; Figure 3). Recurrent glioblastomas had a 37.33% (95% CI: 20.68–53.97%) and a 53.74% (95% CI: 8.69–98.80%) 1-year survival rate for all and for the studies conducted after 2008, respectively, while the single study investigating the effect of mEHT in newly diagnosed tumors [51] reported a 73.33% (95% CI: 57.51–89.16%; vs. all studies: *p* = 0.0021; vs. studies after 2008: *p* = 0.4214) 1-year survival rate. Three studies [43,45,47] investigated whether patients under or over 50 years of age have better 1-year survival rate, and no difference between these patients could be verified (*p* = 0.1129). No difference was found when the type of study (prospective vs. retrospective; *p* = 0.3552), the type of device used during the study (*p* = 0.4273) or the median age of patients (*p* = 0.6778) was compared.

### 3.2. Studies Investigating Tumor Treating Flields in Glioblastoma

The electronic database searches for studies about TTF in glioblastoma patients resulted in a total of 6036 publications. After the removal of duplicates, non-human studies including animal and cellular research reports, reviews and meeting/conference abstracts, 323 articles remained for title and abstract screening. Then, 278 articles were excluded because they either reported results from other tumors, presented results from an animal and/or cellular experiment, were unavailable or had different study interests than the current meta-analysis (e.g., health-economy). Of the remaining 45 studies considered for full text assessment, 18 further studies were removed because they did not include the target variable of this meta-analysis. Two, five and nine articles reported results about the SPARE [52,53], EF-11 [15,16,54,55,56] and EF-14 [17,18,19,57,58,59,60,61,62] studies, of which only one–one was used for the meta-analysis, resulting in a total of 14 available studies to be included in the analyses (Figure 4).

Details of the fourteen TTF studies [15,21,22,26,53,57,63,64,65,66,67,68,69,70] selected for analysis can be read in Table 2. It has to be noted that the EF-11 study results were gathered from the updated post hoc analysis of Kanner et al. [15] instead of from the original [54], because none of those patients who did not finish at least one cycle of therapy were removed from the original publication, causing a significant change in true survival results. A comparison of TTF treatment to a control group was present in five of eleven studies [15,21,57,69,70].

The total number of patients investigating the effect of TTF in glioblastoma was 1309, of which 536 patients (40.9%) died during the first year after study inclusion. A 66.65% pooled 1-year survival rate (95% CI: 52.65–80.65%) was observed for the total cohort receiving TTF, regardless of other clinical parameters. Similar to that of the mEHT results, high heterogeneity [96.5% (95% CI: 95.3–97.4%)] and no publication bias (*p* = 0.6652) was found for the TTF study results. The analysis to identify possible confounding effects revealed 1-year survival rates of 49.01% (95% CI: 1.75–96.27%), 66.29% (95% CI: 48.31–84.27%) and 73.11% (95% CI: 48.89–97.34%) in RCTs, prospective and retrospective studies (*p* = 0.6680), respectively. The effect of when TTF was introduced during the glioblastoma treatment was also investigated: a significantly better 1-year survival rate was found in those patients with a newly diagnosed tumor [82.61% (95% CI: 73.20–92.02%)], compared with those with recurrent tumors [51.74% (95% CI: 30.84–72.64%); *p* = 0.0083; Figure 5].

We were also able to compare the survival rates of 2 and 3 years for eleven and eight studies, respectively: 38.87% (95% CI: 21.73–56.01%) and 34.19% (95% CI: 13.33–55.04%) survival rates were estimated. Neither the year of publication (*p* = 0.9755), the median age (*p* = 0.2682) nor the study type (*p* = 0.7085) affected the 2-year survival rates, but the same difference between recurrent and newly diagnosed glioblastoma was observable (newly diagnosed glioblastoma: 59.79%, 95% CI: 34.40–85.17%; recurrent glioblastoma: 20.18%, 95% CI: 8.18–32.18%; *p* = 0.0057; Figure 6) as described for the 1-year survival rates above. When investigating the 3-year survival rates, patients with newly diagnosed tumors [47.24% (95% CI: 18.31–76.16%)] benefited significantly more from the TTF treatment than those who received TTF for recurrent glioblastoma [11.00% (95% CI: 4.75–17.26%); *p* = 0.0164; Figure 7]. No difference in 3-year survival rates could be justified for the different study types (*p* = 0.2075), years of publication (*p* = 0.4123) or median ages of patients (*p* = 0.0935).

Only a limited number of the available studies (*n* = 5) investigated the effect of TTF over a control cohort. It was found that patients on the TTF-treatment arm had significantly better 1-year [risk ratio (RR): 0.6481, 95% CI: 0.4345–0.9668; *p* = 0.0335; Figure 8A] and 3-year (RR: 0.9215, 95% CI: 0.8819–0.9628; *p* = 0.0003; Figure 8C) survival rates. However, no difference could be observed in the 2-year survival rates of the patients treated with or without TTF (RR: 0.9032, 95% CI: 0.7713–1.0576; *p* = 0.2062; Figure 8B).

### 3.3. The Direct Comparison of Modulated Electro-Hyperthermia and Tumor Treating Flields Studies

We also examined whether there was a difference in the 1-year survival of the patients by directly comparing the mEHT and TTF techniques. It has to be highlighted though that while the majority of the TTF studies were conducted in the last decade, half of the mEHT studies were done prior the general acceptance and use of the Stupp protocol [10,71]. Due to the former and to the fact that glioblastoma survival has significantly improved over the last decade [72], we compared those mEHT studies only with TTF that were performed after 2008. The 1-year survival rate of the 100 and 1289 glioblastoma patients treated with mEHT and TTF was 60.63% (95% CI: 32.21–89.05%) and 63.56% (95% CI: 48.50–78.62%; *p* = 0.8583), respectively. The same results were obtained if the two devices were compared in newly diagnosed glioblastoma (mEHT: 73.33%, 95% CI: 57.51–89.16%; TTF: 79.81%, 95% CI: 70.97–88.65%; *p* = 0.4836; Figure 9) and in recurrent glioblastoma (mEHT: 53.74%, 95% CI: 8.69–98.80%; TTF: 49.24%, 95% CI: 25.90–72.57%; *p* = 0.8618; Figure 10).

## 4. Discussion

Glioblastoma is a highly aggressive tumor with a 5-year survival rate of 1–5% [73]. Its standard treatment includes surgery (if feasible) and radiation therapy with concomitant/adjuvant chemotherapy: procarbazine, lomustine and vincristine (PCV) and temozolomide in the early and late stages, respectively [9,10,74]. Lately, the importance of molecular markers has also emerged [75], e.g., one of the bases of the latest WHO classification of gliomas is the IDH mutation [6]. Additionally, in the last decade an emerging number of reports came to light that the addition of non-invasive, device-based concomitant therapies, mEHT or TTF, might further increase therapy response. Moreover, several studies reported that even if no large differences in patient survivals could be justified, the quality of life of patients was much higher compared with those without the additional treatment options [27,34,76].

A less than 20% 1-year survival rate was reported approximately twenty years ago [77], which has almost doubled today [13,15], moreover, some studies achieved 1-year survival rates over 80% [11], but still, only modest median overall survivals can be achieved [13,15]. If concomitant mEHT or TTF was added to the treatment plan, an average 42% and 64% 1-year survival rate could be achieved, respectively. Similar to the trend reported in the meta-analysis of Poon et al. [72], we observed that the 1-year survival rate significantly improved in those mEHT studies, which were conducted later, than 2008. Taking into account this observation, the adjusted 1-year survival rates were 61% and 67% for the mEHT and TTF studies, respectively, both of which are significantly greater compared with those observed in patients treated without the devices [12]. Survival rates for longer intervals were only available for the TTF studies, and a 39% 2-year and a 34% 3-year survival rate was found. For comparison, by treating glioblastoma patients with either the standard (60 Gy irradiation + 6 cycles of temozolomide) or the extended (temozolomide cycles > 6) Stupp protocol, the reported 2-year and 3-year survival rates are lower [12,71,78], but the opposite was also reported in another study [11].

As a somewhat expected result, further findings of the current meta-analysis included that newly diagnosed glioblastoma patients can benefit more from the early use of TTF: a 83% vs. 52% 1-year survival rate was found for the newly diagnosed and recurrent glioblastoma patients, respectively, although some authors assumed the exact opposite [69]. Similar results could be obtained in the case of mEHT (73% vs. 54%), however, only one study investigated the effect of mEHT in newly diagnosed glioblastoma patients [51], which immediately raises the need for additional studies investigating the effect of mEHT in this setting.

A few studies also investigated whether the use of either of the two devices has a significant advantage compared with conventional treatment. While in the case of TTF we managed to identify five studies that compared patients treated with and without TTF, only one mEHT study made a similar comparison. It has to be noted that the Fiorentini study [46] tested the palliative use of mEHT vs. other palliative options only. For TTF, we found significantly reduced 1- and 3-year mortality rates in the treated with TTF groups, compared with those without TTF. For mEHT, Fiorentini et al. [46] have described significantly longer overall survival times in the mEHT-treated group.

By examining the details of the available clinical trial results, the following can be further confirmed about the two devices. With the introduction of mEHT in the treatment plan, several studies could report improved responses to the treatment [23,44,45,46,47,48], a better quality of life [44,46], increased functional activity of patients measured by the Karnofsky Performance Score scale [51] and complete and/or partial response could be maintained in some cases for longer periods of time as well [45,46,47,48,49,51]. It has to be noted, however, the result on age response is controversial: Fiorentini et al. [46] found no difference between the survival of patients over or under the age of 50 years, while Roussakow [44] and Sahinbas et al. [43] have found the opposite. In this study, we could not justify difference between the younger and older cohorts. Furthermore, in the Phase I study of Wismeth et al. [23] it has been reported that at least three mEHT treatments per week are required for an effective response. As with other treatments options, a few complications of mEHT have been confirmed. Basically, all of the mEHT studies reported only grade I and II side effects: headaches, skin redness and/or mild burning at the treatment site, nausea and vomiting, dizziness, neurological symptoms (aphasia, seizures) and grade I/II anemia and/or leukopenia and/or thrombocytopenia [23,43,44,45,46,47,48,50,51].

Most results about TTF are known from the EF-11 [15,16,54,55,56] and EF-14 [17,18,19,57,58,59,60,61,62] randomized trials. The first study has investigated 120 chemotherapy-free, only TTF-treated and 117 control patients receiving active chemotherapy, and it could report a marginal difference in survival only [54]. However, by further analyzing the study results [15]—by excluding patients who did not finish at least a single cycle of therapy—the therapeutic advantage of TTF over patient survival became verifiable. In contrast, the concomitant effect of TTF over chemotherapy (first-line: temozolomide) was investigated in the EF-14 study; 466 and 229 patients were treated with and without TTF, respectively [57]. Similar to that of the results of the mEHT studies, a more durable complete and/or partial response to therapy and/or stable disease was more common in the TTF-treated groups [16,26,54,55,62], and TTF-treated patients had a better overall and progression-free survival [18,19,21,22,26,54,57,58,59,62,63,64,67,68,70] and stable or improved quality of life status (except for itchy skin [19,60,61,79]) [19,53,54,60,61]. A better compliance to the treatment can improve treatment and prolong survival time [15,16,17,18,19,20,21,22], TTF plus chemotherapy was superior in all age groups compared with chemotherapy alone [19,57], TTF alone is superior to bevacizumab-only chemotherapy [16] and its efficacy might be further improved if 6-thioguanine, lomustine, capecitabine and celecoxib (TCCC) is in combination with bevacizumab [64]. A higher local minimum filed intensity, power density and dose density of the TTF-device is associated with better overall and progression-free survival [17], moreover, no difference has been reported in the cognitive status changes between patients treated with or without TTF [60]. In addition to the EF-11 and EF-14 study result findings, it has been reported that patients with *PTEN* (phosphatase and tensin homolog) mutations have longer survival compared with those with wild type *PTEN* (22 months vs. 12 months) [70]. TTF after skull remodeling surgery is safe and a positive correlation between field enhancements and burr hole sizes with a plateau at 15–20 cm^2^ has been described [67]. A triple-drug regimen of temozolomide, bevacizumab and irinotecan with TTF is superior to other bevacizumab-based chemotherapies with TTF [65]. Scalp sparing radiation with concurrent temozolomide and TTF is well tolerated by patients, furthermore, a better response for glioblastoma patients with methylated O^6^-methylguanine DNA methyltransferase (MGMT) promoter has been observed [52,53,80]. In contrast, *IDH1* and/or *IDH2* mutation status has not affected the survival of patients [80]. Mutations in the phosphatidylinositol-4,5-bisphosphate 3-kinase catalytic subunit alpha and epidermal growth factor receptor genes were associated with a decreased or no response to TTF, while the mutated neurofibromatosis type 1 gene has been associated with better overall and progression-free survival, and the tumor protein p53 gene mutations have had no effect on any outcome upon TTF therapy [21]. The following common side effects of TTF have been observed: mild to moderate contact dermatitis (“medical device site reaction beneath the transducer arrays”), headache, fatigue, convulsion or seizure, confusion, mental status changes, mild anemia and/or lymphopenia and/or thrombocytopenia, diarrhea or constipation and neurological decompensation [19,22,26,52,53,54,57,58,59,62,63,64,65,66,67,79]. Further results are expected after the completion of the following TTF clinical trials: NCT05310448, NCT04223999, NCT03642080, NCT04469075, NCT04474353, NCT03477110, NCT04689087, NCT04471844, NCT04397679, NCT04671459, NCT04421378, NCT04492163, ChiCTR2100047049, ChiCTR2100041969, JPRN-UMIN000041745 and ISRCTN14267833.

To our knowledge, this is the first study that systematically reviewed the current knowledge about mEHT in glioblastoma and compared mEHT and TTF directly. In relation to TTF, several reviews and a few meta-analyses have already been published, which in certain aspects are much more detailed than the present work, including but not limited to [27,81,82,83,84,85]. In the meta-analysis of Regev et al. [81], the pooled 1-, 2- and 3-year survival rates were 73%, 45% and 29%, respectively, which are comparable to the ones calculated in the current analysis (67%, 39% and 34%). Similarly, the pooled 1-year survival rate of 47.3% reported by Li et al. [82] for recurrent glioblastoma is not different from the 52% we observed.

### Strength and Limitations of the Study

To the best of our knowledge, we are the first to analyze mEHT and TTF study results in a single meta-analysis. However, a few limitations of this analysis should be mentioned, including that only a limited number of trials could be investigated and most of them were non-randomized trials. Although the number of studies analyzed in the meta-analysis could have been slightly increased if median overall/progression-free survivals were used instead of the x-year survival rate, their calculation would have given inaccurate results [86]. Heterogeneity of the included studies was high, which might have also introduced some bias. Another limitation of the current study was that the number of available studies investigating mEHT and TTF were significantly different, which is due to the fact that, up to now, the number of centers adopting mEHT is very limited around the world, as opposed to TTF. This ultimately might have affected the calculated pooled effect sizes.

## 5. Conclusions

In conclusion, this study investigated the beneficial effects of (concomitant) mEHT and TTF over conventional chemoradiotherapy in glioblastoma. It was found that both mEHT and TTF could significantly increase the survival of glioblastoma patients and the same survival rates can be achieved using both devices in the cohorts of newly diagnosed and recurrent glioblastomas. It has to be emphasized, however, that the small number of centers using mEHT largely limits its application, and there is no data about the combined use of the two devices, therefore, further studies are recommended.

## Figures and Tables

**Figure 1 cancers-15-00880-f001:**
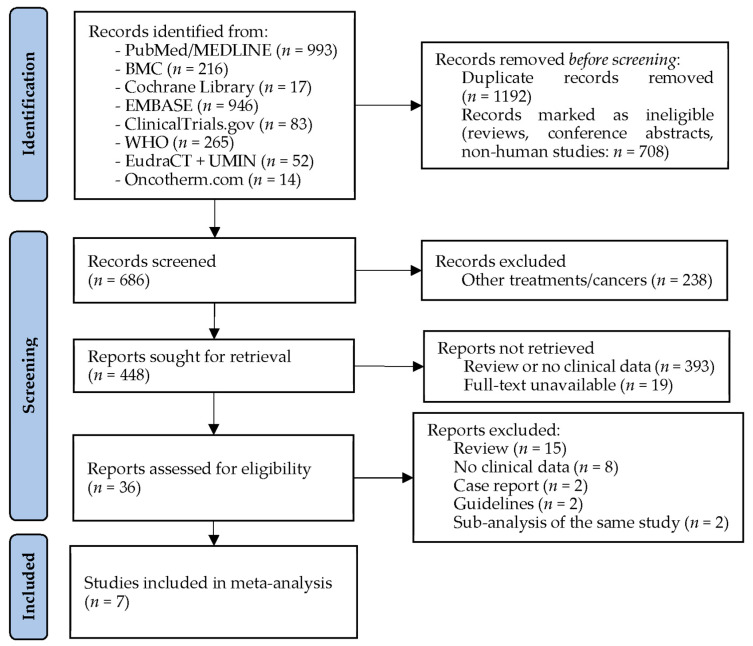
PRISMA flow diagram of studies about modulated electro-hyperthermia. BMC: BioMed Central; EMBASE: Excerpta Medica Database; EudraCT: European Union Drug Regulating Authorities Clinical Trials Database; MEDLINE: Medical Literature Analysis and Retrieval System Online; WHO: World Health Organization’s International Clinical Trials Registry Platform; UMIN: University hospital Medical Information Network Clinical Trials Registry (for Japan).

**Figure 2 cancers-15-00880-f002:**
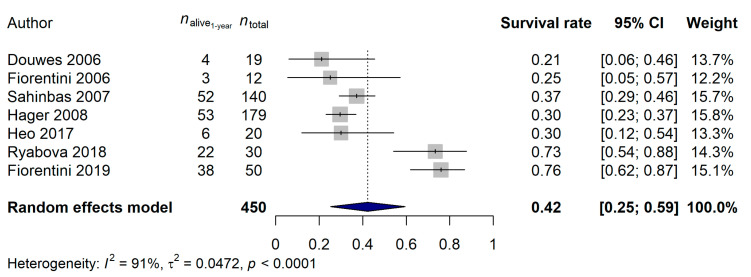
Effect of modulated electro-hyperthermia on 1-year glioblastoma survival rate [43,46,47,48,49,50,51].

**Figure 3 cancers-15-00880-f003:**
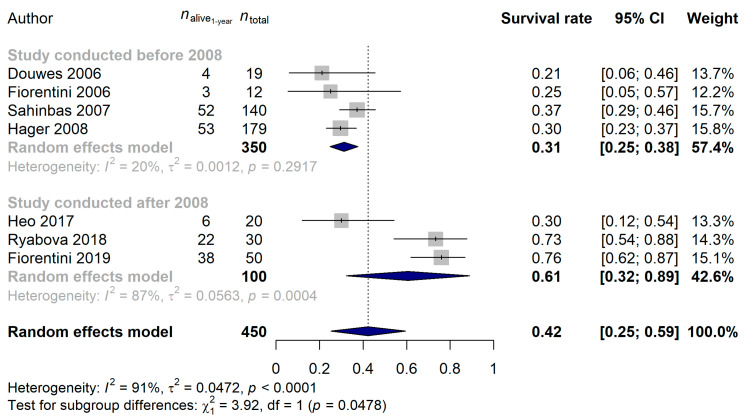
Effect of modulated electro-hyperthermia on 1-year glioblastoma survival rate, grouped by studies published before and after 2008 [43,46,47,48,49,50,51].

**Figure 4 cancers-15-00880-f004:**
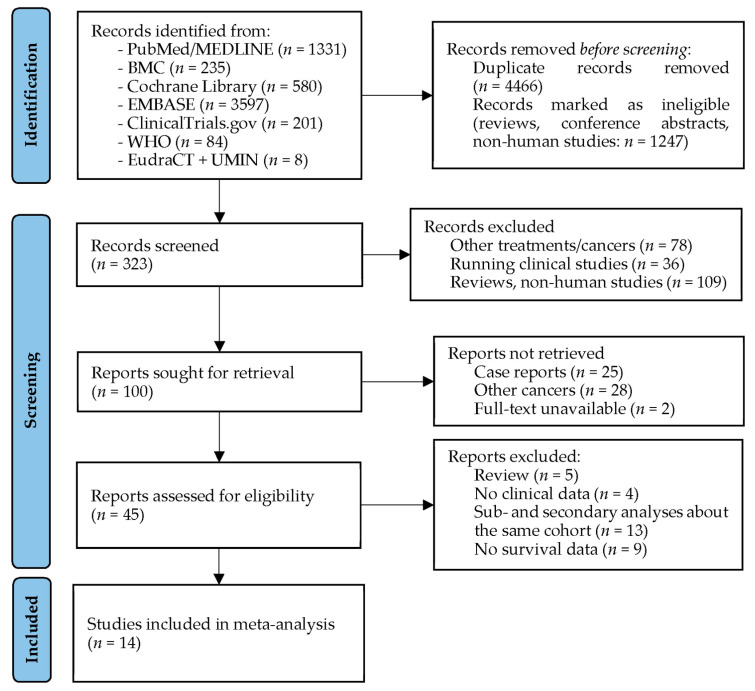
PRISMA flow diagram of studies about Tumor Treating Fields. BMC: BioMed Central; EMBASE: Excerpta Medica Database; EudraCT: European Union Drug Regulating Authorities Clinical Trials Database; MEDLINE: Medical Literature Analysis and Retrieval System Online; WHO: World Health Organization’s International Clinical Trials Registry Platform; UMIN: University hospital Medical Information Network Clinical Trials Registry (for Japan).

**Figure 5 cancers-15-00880-f005:**
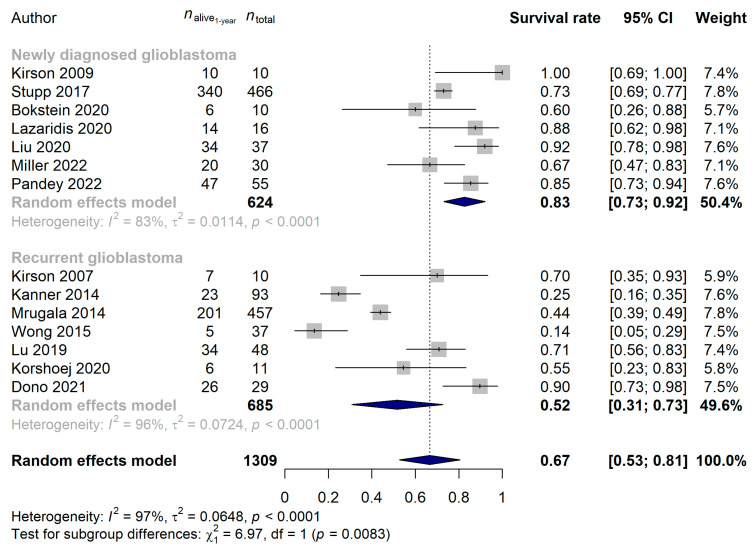
Significantly better 1-year survival rates were found when the Tumor Treating Fields treatment was introduced at an earlier stage of treatment (*p* = 0.0083) [15,21,22,26,53,57,63,64,65,66,67,68,69,70].

**Figure 6 cancers-15-00880-f006:**
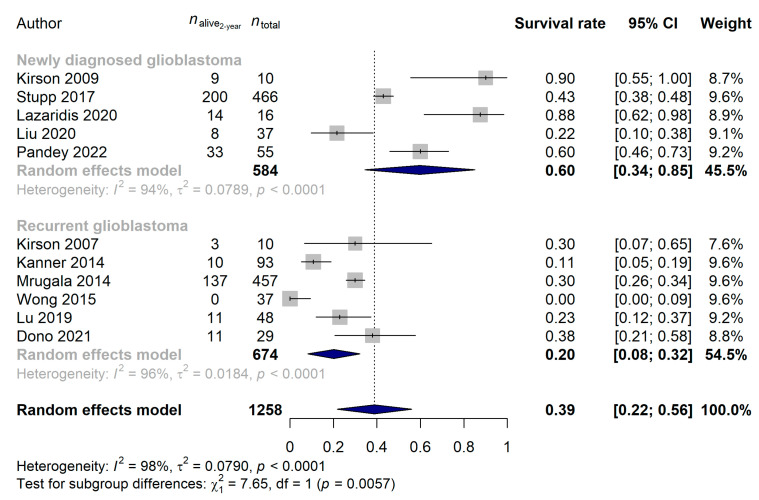
Significantly better 2-year survival rates were found when the Tumor Treating Fields treatment was introduced at an earlier stage of treatment (*p* = 0.0057) [15,21,22,26,57,63,64,65,68,69,70].

**Figure 7 cancers-15-00880-f007:**
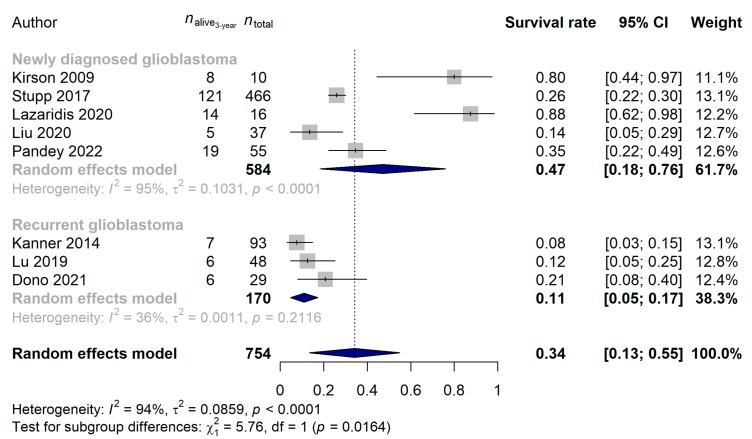
Significantly better 3-year survival rates were found when the Tumor Treating Fields treatment was introduced at an earlier stage of treatment (*p* = 0.0164) [15,21,57,63,65,68,69,70].

**Figure 8 cancers-15-00880-f008:**
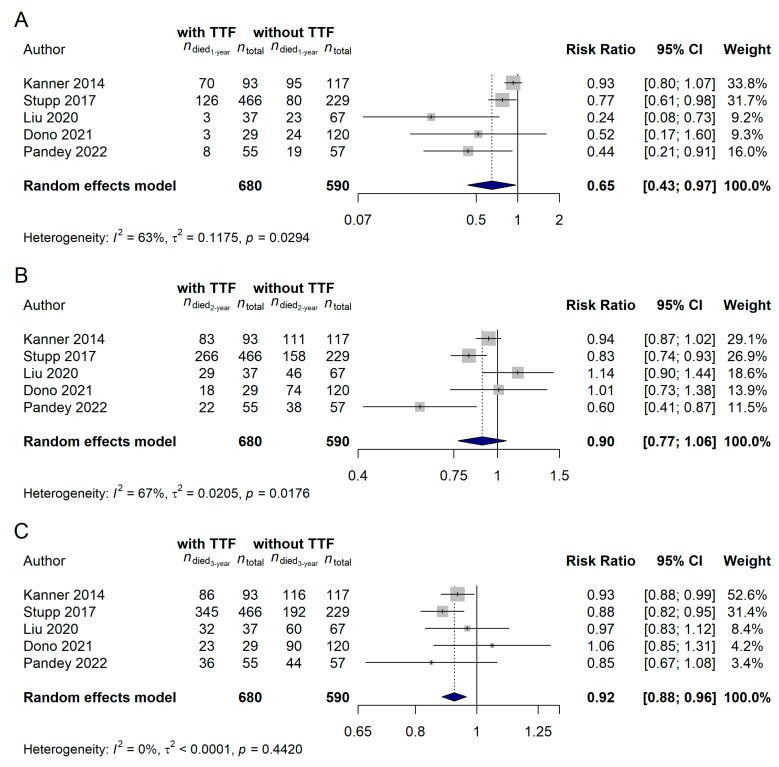
Glioblastoma patients treated with Tumor Treating Fields (TTF) had significantly better (**A**) 1-year (*p* = 0.0335) and (**C**) 3-year (*p* = 0.0003) survival rates, while no difference in the (**B**) 2-year survival rates could be justified (*p* = 0.2062) compared with those who did not receive TTF during their treatment [15,21,57,69,70].

**Figure 9 cancers-15-00880-f009:**
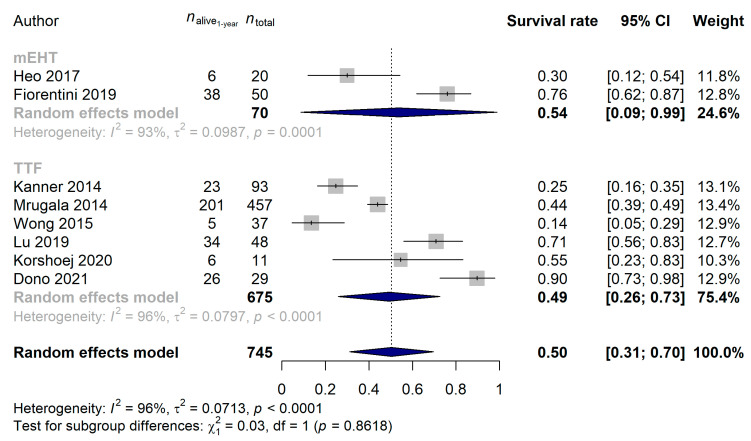
In newly diagnosed glioblastoma, the non-inferiority (*p* = 0.4836) of modulated electro-hyperthermia (mEHT) could be observed compared with the more widely applied Tumor Treating Fields (TTF). It has to be noted that due to the wider acceptance of the Stupp protocol [10,71] and that patient survival significantly improved in the last decade [72], only the comparison of studies conducted after 2010 were compared [15,22,46,50,64,65,67,70].

**Figure 10 cancers-15-00880-f010:**
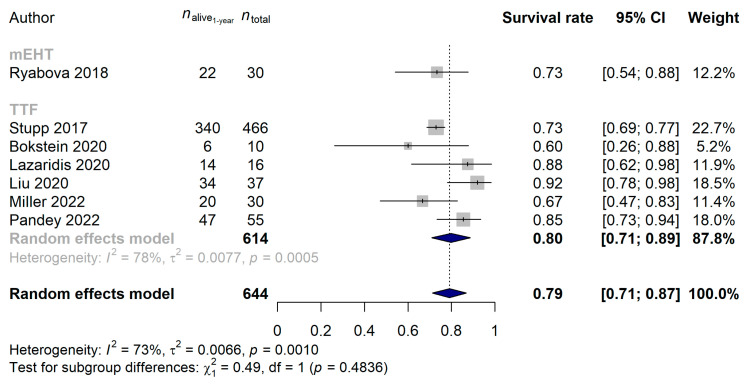
In recurrent glioblastoma, the non-inferiority (*p* = 0.8618) of modulated electro-hyperthermia (mEHT) could be observed compared with the more widely applied Tumor Treating Fields (TTF). It has to be noted that due to the wider acceptance of the Stupp protocol [10,71] and that patient survival significantly improved in the last decade [72], only the comparison of studies conducted after 2010 were compared [21,51,53,57,66,68,69].

**Table 1 cancers-15-00880-t001:** Details of the selected studies investigating the effect of modulated electro-hyperthermia (mEHT) in glioblastoma.

Author (Year)	Type of Study	Cases (*n*)	mEHT Device	Additional Therapy	Age (Median)	Females
Douwes et al. (2006) [47]	Prospective	19	Oncotherm EHY2000	nimustine	55	– ^1^
Fiorentini et al. (2006) [48]	Prospective	12	Oncotherm EHY2000	– ^1^	– ^1^	– ^1^
Sahinbas et al. (2007) [43]	Retrospective	140	Oncotherm EHY2000	temozolomide and/or herbal medicines and/or irradiation	44	35.7%
Hager et al. (2008) [49]	Prospective	179	LRF-DHT	– ^1^	– ^1^	– ^1^
Heo et al. (2017) [50]	Prospective	20	Celsius 42+	re-irradiation	56	60%
Ryabova et al. (2018) [51]	Prospective	30	Celsius 42+	temozolomide + irradiation	56	36.7%
Fiorentini et al. (2019) [46]	Retrospective	50	Oncotherm EHY2000	no	55	– ^1^

^1^ Not detailed in the original article. LRF-DHT: deep hyperthermia with low radiofrequency capacitive coupled electrodes.

**Table 2 cancers-15-00880-t002:** Details of the selected studies investigating the effect of Tumor Treating Fields (TTF) in glioblastoma.

Author (Year)	Type of Study	Cases (*n*)	Controls (*n*)	Additional Therapy	Age (Median)	Females
Kirson et al. (2007) [26]	Prospective	10	–	temozolomide	– ^1^	– ^1^
Kirson et al. (2009) [63]	Prospective	10	–	temozolomide	– ^1^	– ^1^
Kanner et al. (2014) [15]	RCT	93	117	no	54	– ^1^
Mrugala et al. (2014) [22]	Prospective	457	–	no restriction on any combination therapies, but not detailed	55	32.4%
Wong et al. (2015) [64]	Retrospective	37	–	bevacizumab with or without 6-thioguanine, lomustine, capecitabine, and celecoxib (TCCC)	57	37.8%
Stupp et al. (2017) [57]	RCT	466	229	temozolomide	56	32.2%
Lu et al. (2019) [65]	Retrospective	48	–	temozolomide + bevacizumab + irinotecan or bevacizumab-based chemo regimen	55	33.3%
Bokstein et al. (2020) [66]	Prospective	10	–	temozolomide + irradiation	60	20%
Korshoej et al. (2020) [67]	Prospective	11	–	chemotherapy after skull remodeling surgery	57	18.2%
Lazaridis et al. (2020) [68]	Retrospective	16	–	lomustine + temozolomide	50	43.8%
Liu et al. (2020) [69]	Retrospective	37	67	temozolomide + irradiation	61	37.8%
Dono et al. (2021) [70]	Retrospective	29	120	temozolomide + irradiation	58	34.5%
Miller et al. (2022) [53]	Prospective	30	–	temozolomide + irradiation	58	33.3%
Pandey et al. (2022) [21]	Retrospective	55	57	temozolomide	59	30.9%

^1^ Not detailed in the original article. RCT: randomized clinical trial.

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
