# Peer review of "Meta-Analysis of Modulated Electro-Hyperthermia and Tumor Treating Fields in the Treatment of Glioblastomas"

_cancers, 2023, doi:10.3390/cancers15030880_

Round 1

Reviewer 1 Report

The authors present a review and metaanalysis of mEHT and TTFields. The review is relevant and provides and important perspective on two electric treatments, of which one (TTFields) is widely adopted. Althoug interesting, well-written and reasonably well-designed, I have a number of concerns about the design and methodology in the study, that I believe should be addressed. 

Minor comments:

1) Glioblastomas are not astrocytomas grade IV. Ther former are IDH-wildtype, while the latter are IDH mutated. This should be corrected in the introduction (WHO 2021 classification)

Major comments:

2) The authors used the search terms "tumor-treating fields". However, it is common to denote the treatment "tumor treating fields" or "TTFields" or "alternating electric fields". The authors should include these terms and redo their analysis. 

3) The authors do not state specific inclusion- or exclusion criteria for studies in the search. This should be stated and related directly to the inclusion chart. 

4) The authors used automatized tools for eligibility assessment, but it is not described in the methods section and the criteria for evaluation are not stated. 

5) The authors do not state whether records and articles were reviewed by one or more individuals and how conflicts were resolved. 

6) The authors state that they used the prisma guidelines but several items from the checklist are missing. 

7) In fig five the authors present a collective analysis of all TTFields trials regardless of new GBM diagnosis og recurrent. This is not a relevant analysis in my opinion, as it these patients populations behave very differently with respect to the investigated outcome. The analysis should be separated as done later in the tekst. 

8) TTFields has been investigated in many reviews and recent metaanalyses. The authors should argue how their study is novel and how the results compare more extensively to recent similar analyses.

9) The authors compare mEHT and TTFields within the last decade, when Stupp regime was widely used, which is relevant. The authors should further homogenize their analyses to make sure the comparison is fair and clinically relevant. This would also include separating diagnoses into newly diagnosed and recurrent disease. 

Reviewer 2 Report

in order to obtain a satisfying minor revision I think it could be sufficient to modify the paragraph from line 51 to line 54, introducing the idea that an other device is able to treat brain tumors, such as the Nanotherm therapy of Magforce AG.  

Authors can change the introduction (line 51) “ in the last two decades , three (instead of  “two”) electromagnetic devices/techniques ………

In the  line 53 Authors, after the listing of mEHT and TTF,  have to include a brief description of Magforce System explaining the mechanism of Magnetic Hyperthermia.

The same device has to be cited  in the Abstract and in the References list.

Without  this minor revision the manuscript could be considered incomplete and the review not truthful. In my opinion , with this addition  the article is worthy of being published.

Reviewer 3 Report

The authors present an interesting manuscript, that is relevant to the field. Some aspects should be considered before publication:

-When addressing survival data, the authors quote historic outcomes. Novel publications such as Herrlinger et al 2019 have achieved far better outcomes even in control populations

-Fig 1 one number is missing in the "excluded reports" box

-When discussing outcomes/survival in the discussion, the authors again should mention more recent better outcomes that have substantially improved. Further, they might mention the importance of molecular markers in this context

-The conclusion should be revised substantially: the authors should refrain from quoting other manuscripts in their conclusion, they should not ask (rhetorical) questions and most importantly should limit the conclusion to claims substantiated by their data

-the authors mention immune responses for the first time in the conclusion and should delete this section from the conclusion

Round 2

Reviewer 1 Report

None

Reviewer 3 Report

The authors have sufficiently revised their manuscript.